# Salt, Not Always a Cardiovascular Enemy? A Mini-Review and Modern Perspective

**DOI:** 10.3390/medicina58091175

**Published:** 2022-08-29

**Authors:** Mihai Hogas, Cristian Statescu, Manuela Padurariu, Alin Ciobica, Stefana Catalina Bilha, Anca Haisan, Daniel Timofte, Simona Hogas

**Affiliations:** 1Physiology Department, “Grigore T. Popa” University of Medicine and Pharmacy, Universitatii 16, 700115 Iasi, Romania; 2Cardiology Department, “Grigore T. Popa” University of Medicine and Pharmacy, Universitatii 16, 700115 Iasi, Romania; 3Psychiatry Department, “Grigore T. Popa” University of Medicine and Pharmacy, Universitatii 16, 700115 Iasi, Romania; 4Department of Biology, Faculty of Biology, Alexandru Ioan Cuza University, B dul Carol I, No 11, 700115 Iasi, Romania; 5Academy of Romanian Scientists, Splaiul Independentei Nr. 54, Sector 5, 050094 Bucuresti, Romania; 6Center of Biomedical Research, Romanian Academy, B dul Carol I, No 8, 700115 Iasi, Romania; 7Endocrinology Department, “Grigore T. Popa” University of Medicine and Pharmacy, 700115 Iasi, Romania; 8Surgery Department, “Grigore T. Popa” University of Medicine and Pharmacy, 700115 Iasi, Romania; 9Nephrology Department, “Grigore T. Popa” University of Medicine and Pharmacy, 700115 Iasi, Romania

**Keywords:** salt, salt-sensitive hypertension, salt restriction

## Abstract

Dietary salt intake is a long-debated issue. Increased sodium intake is associated with high blood pressure, leading to salt-sensitive hypertension. Excessive salt intake leads to arterial stiffness in susceptible individuals via impaired nitric oxide action and increased endothelin-1 expression, overactivity of the renal sympathetic nervous system and also via aldosterone-independent activation of the mineralocorticoid receptor. Salt restriction in such individuals reduces blood pressure (BP) values. The optimal level of salt restriction that leads to improved cardiovascular outcomes is still under debate. Current BP and dietary guidelines recommend low sodium intake for the general population. However, a specific category of patients does not develop arterial hypertension in response to sodium loading. In addition, recent research demonstrates the deleterious effects of aggressive sodium restriction, even in heart failure patients. This mini review discusses current literature data regarding the advantages and disadvantages of salt restriction and how it impacts the overall health status.

## 1. Introduction

The subject of dietary salt has for a long time been a focus of clinical research, and still represents a subject of intense study and debate. The special interest on dietary salt arises from the fact that sodium could represent a relatively easy to control health risk factor, among other exogenous risk factors such as tobacco, alcohol, or caloric intake [1].

In this way, there is a strong belief that high salt consumption is involved in several health problems including high blood pressure (BP), stroke, left ventricular hypertrophy, heart failure, renal disease, renal lithiasis and also gastric cancer [2,3]. Indeed, salt restriction is associated with several health benefits in the general population, such as reduced incidence of arterial hypertension, kidney disease and, hence, improvement in disability-adjusted life years (DALYs) [4,5].

Special attention is given to the direct implication of salt intake in various cardiovascular diseases (CVD). Evidence of the salt hypothesis shows that reducing the amount of salt intake with as little as 1.8 to 2.5 g/day will result in a decrease of BP levels and in a subsequent reduction of cardiovascular events [6]. This applies to salt-sensitive hypertension, which is defined as the increase in BP values by at least 5% secondary to salt-loading [7,8]. However, there is an inter-individual variation to the blood pressure raising effects of sodium charge, with a specific category of patients that do not develop significantly increased BP values with increased salt intake, and thus are termed to have salt-resistant hypertension [8,9]. 

In this context, there is growing scientific evidence sustaining health benefits of dietary salt restriction, with several BP guidelines and most dietary guidelines recommending low sodium intake (<2.3 g Na/day) for the general population [10]. However, recent research in the field is controversial, with data pointing towards deleterious effects of severe sodium restriction in the general population. Optimal sodium intake has also been challenged in patients suffering from heart failure (HF). This narrative mini-review discusses current literature data regarding the advantages and disadvantages of salt restriction in the general population from the health point of view. Knowledge gaps and up-to-date recommended interventions are presented. The latest approaches in particularly fragile situations such as HF or chronic kidney disease (CKD) are also reviewed. 

## 2. Methods

We searched the electronic database of PubMed and ISI Web of Science from inception until April 2022 using the keywords “dietary sodium”, “low-salt diet”/“low-sodium diet”, “arterial hypertension”, “cardiovascular diseases”, “heart failure” and “chronic kidney disease” in any combination. Meta-analyses, randomized controlled trials, observational studies and guidelines were included if the relationship between sodium intake and CVD, cardiovascular (CV) events or mortality was referred to in the general population; data regarding sodium intake in HF and in CKD patients, respectively, were considered if highly relevant. Relevant references from the selected articles and guidelines were also searched manually. 

## 3. Discussion

### 3.1. Current Recommendations for Salt Intake

#### 3.1.1. Salt Requirement Recommendations for the General Population

The current dietary health recommendation is to decrease the populational level of salt consumption from 9–12 g/day, which is excessive from an evolutionary point of view, to a level of below 5 g/day salt intake or even more [4,11]. In fact, the World Health Organization recommends a maximum adult salt intake of 5 g/day (2 g sodium/day) [11], while The American Heart Association recommends a total sodium intake between <2.3 g/day (equivalent to 5.7 g salt/day) to <1.5 g/day (equivalent to 3.75 g salt/day) in high risk-patients for CVD [12]. 

However, an aggressive sodium restriction still hasn’t proven to be effective in reducing the risk of cardiovascular events overall [10]. In a recent statement endorsed by the European Society of Cardiology, O’Donnell et al. [10] argue that most of the global population follows a moderate sodium-diet (between 2.3 and 4.6 g/day) that is not associated with an increased cardiovascular risk, and that only exceeding 5 g/day (12.5 g salt/day) sodium requires dietary restrictions.

#### 3.1.2. Salt Restriction Recommendations in Particular Situations: HF and CKD

An extreme sodium restriction is discouraged, since it was proven that a very low-sodium diet is associated with negative health consequences due to competing “off-target” adverse consequences upon activation of the sympathetic response and renin-angiotensin-aldosterone (RAS) system, which increased the risk for cardiovascular events [10]. This led to paradigm shifts in the official recommendations for sodium intake in HF patients, where the latest American (2022) and European (2021) guidelines recommend avoiding excessive salt intake, defined as more than 5 g of salt per day [13,14]. As a more consistent reduction in sodium intake close to the lower range suggested by the American Heart Association is difficult to achieve and sustain in the long-term, even small decrements in diet sodium it may still be beneficial in particular cases, such in as salt-sensitive hypertensive patients [12]. 

Regarding the recommendations for salt intake in patients with chronic kidney disease (CKD), the latest Clinical Practice Guideline for the Management of Blood Pressure in CKD ((KDIG) 2021) promotes sodium restriction to less than 2 g/day (5 g salt/day) in patients with high BP and CKD [15].

### 3.2. Salt-Sensitive versus Salt-Resistant Hypertension

Guyton [16] was the first to propose a model of salt-sensitive hypertension, according to which kidney pressure natriuresis impairment is required in order for arterial hypertension to develop (“the natriuretic dysfunction theory”). The classic Guytonian model of salt-sensitive hypertension has recently been challenged by the renal vaso-dysfunction theory [17]: according to Kurtz et al. [18], it is the abnormal renal vascular resistance that initiates salt-induced high BP. 

Indeed, salt-sensitive hypertension is associated with nitric oxide (NO) synthase polymorphism in susceptible individuals [19]. Salt-induced impairment of NO synthase promotes renal vascular dysfunction, with initiation of hypertension [20]. Moreover, increases in plasma sodium are associated with endothelial cell stiffness, with reduced NO bioavailability [18,21,22]. 

Thus, an important structure that mediates the effects of sodium in hypertension is represented by the endothelium. It seems that the relation between the endothelium and salt intake is quite complex, while it is known that salt ingestion is activating the endothelium cells. This led to a shift of the paradigm from impaired renal sodium excretion towards systemic vascular dysfunction (Figure 1) [23].

The endothelium is not a simple detector of sodium concentrations, but it also reacts and suffers transformation in relation to salt intake. The endothelial cells respond to salt intake through cellular signaling, which involves production of TGF-β_1_ and nitric oxide as mediators [24]. The increased production of TGF-β_1_ promotes peripheral arterial constriction, which in turn leads to increased BP [25]. Salt-sensitive individuals largely lack the normal vasodilator response secondary to increased cardiac output induces by salt loading [23].

Also, the mechanism involved in the endothelial response to sodium load is not necessarily dependent on blood pressure changes due to salt intake: a high-salt diet induces an exaggerated upregulation of the potent vasoconstrictor endothelin-1 expression in salt-sensitive individuals, leading to worse endothelial dysfunction compared to salt-resistant subjects [23].

A long-term high-sodium diet intake could determine endothelial dysfunction and kidney mass reduction, causing a certain vulnerability to salt intake in these individuals. An average reduction of 2 g in daily sodium intake significantly reduces arterial stiffness, according to recent meta-analysis [26]. For instance, a study published in 2011 on young and adult uninephrectomized rats fed with a high-salt diet for four weeks showed a significant increase in BP (180.9 ± 22.6 mmHg) and proteinuria (100.3 ± 44.5 mg/day), compared to normal sodium fed rats (BP 117.8 ± 9.5 mmHg, proteinuria 8.8 ± 1.2 mg/day). In addition, individuals fed a high-salt diet showed lesions in renal glomeruli and interstitial tissue, together with local inflammatory markers, compared to normal sodium fed rats. This further supports evidence of renal injury. The results of the study prove that a high-sodium diet increases BP and has a noxious effect on renal integrity [27]. Data were confirmed in human studies, where salt-sensitivity increases with age [28]. That is why a similar reduction of approximately 1.7 g Na/day leads to a greater average BP reduction in older compared to younger adults (7.5 versus 5.3 mm Hg) [29]. Besides the BP detrimental effects, a long-term salt load is associated with the decline in kidney function in hypertensive patients; also, the estimated glomerular filtration rate (eGFR) is inversely correlated with urinary salt excretion, independently of BP values and antihypertensive medication used [30]. A meta-analysis of randomized controlled trials published in 2015 demonstrated a 32.1% reduction in urinary albumin excretion following an average reduction in sodium intake of 2.1 g, more so in patients with kidney damage [31].

Another important mediator of salt and water intake on blood pressure is the RAS system, which plays a unique hemodynamic role by regulating both vascular reactivity and arterial BP. Given the fact that the activity of the RAS system is dependent on salt consumption, a decrease in salt intake produces an increase in renin and aldosterone levels in order to compensate for a sodium deficit [32]. In addition, the RAS system can increase salt and water conservation by adjusting renal perfusion and glomerular filtration rate via angiotensin II [33]. Conversely, an increase in sodium intake augments renal blood perfusion, decreasing the renal vascular resistance. NO is an essential mediator between salt and the hemodynamic response to water and salt loading. It is also assumed that the mineralocorticoid receptor (MR) activation would be a mediator in kidney injury in young rats that are fed a high salt-diet, and thus the administration of mineralocorticoid antagonist would improve kidney function and also blood pressure [34].

Experimental studies demonstrated an abnormal and aldosterone-independent activation of MR associated with salt-loading via Rac1—a modulator of the MR [35]. Salt loading normally downregulates Rac1 activity, thus decreasing the MR activity and maintaining normal BP. In experimental models of salt-sensitive hypertension, Rac1 is activated by high salt intake. This leads to aldosterone-independent MR activation in the kidney, thus elevating BP and promoting renal injury [9]. Hirohama et al. [36] recently demonstrated that salt-induced activation of Rac1 in the distal nephrons contributes to diabetic kidney disease progression via podocyte injury and elevated BP.

In addition, the renal sympathetic nervous system (SNS) overactivity in response to salt-loading contributes to high BP in salt-sensitive individuals via increased renin secretion, reduced renal blood flow and enhanced sodium tubular reabsorption [9]. In addition, the aldosterone-induced activation of the MR in response to renin secretion promotes the disruption of the glomerular filtration barrier and development of proteinuria in animal models of metabolic syndrome [34].

It is also important to specify that a long-term high sodium intake leads to a decrease in the number of skin capillaries, which actually may represent the first step in developing hypertension [37]. According to the 3-compartment model proposed by Titze et al. [38], sodium storage also takes place in the skin. The skin mimics the countercurrent system in the kidneys and represents an extra-renal mechanism of BP regulation after high-salt intake. The vasodilator response of the skin may be attenuated in predisposed individuals, leading to the development of salt-sensitive hypertension (Figure 1) [39].

### 3.3. Salt and Cardiovascular Events

As hypertension is a risk factor for CVD and sodium restriction lowers BP, one would assume that a low-salt diet would lower the risk for adverse cardiovascular events. Earlier meta-analyses of cohort studies and randomized controlled studies demonstrated an increased risk of stroke, stroke mortality and coronary heart disease mortality associated with increased sodium intake [40,41]. Nonetheless, the range of results regarding the effect of increased diet sodium on all-cause mortality and all CVD is quite wide (Table 1) [40,41,42]. Recently, the theory of a J-shaped relationship between sodium intake and cardiovascular events emerged from large cohort studies [43,44] and was confirmed by meta-analyses. Comparing the usual sodium intake (2.5–7 g/day) with a low-sodium diet (<2.7 g/day) revealed a decreased risk of all-cause mortality and CVD incidence in the usual diet versus sodium restriction [42]. A more recent meta-analysis published by Jayedi et al. [45] confirmed a linear relationship between dietary sodium intake and stroke risk (pooled RR 1.06 for every 1 g/day increment in dietary sodium intake), without any evidence of a J-shaped curve. Thus, current evidence is still limited and interpretation of published data is divergent, while large randomized controlled trials investigating the effect of sodium reduction upon CVD and mortality are still awaited [10].

### 3.4. Low-Salt Diet—Advantages

#### 3.4.1. Low-Salt Diet and BP

Clinical studies have confirmed the BP-lowering effect of diet salt reduction in the elderly. In the *Intersalt Study*, an epidemiological comparative study on 10.079 participants from 52 international centers, diet sodium correlated with an increase in BP when associated with age. It was estimated that lowering the amount of sodium consumption by 100 mmol/day would correspond to an average systolic BP reduction of at least 2.2 mm Hg [46]. These results were also confirmed by similar studies, such as the TOHP Study [6]

Nevertheless, even a modest reduction in salt intake decreased BP levels in the long-term in patients with either normal or increased BP, although the effect seems more important in the latter, according to dedicated meta-analyses [47]. In individuals with high BP, the median decrease in 24-h urinary sodium excretion was 75 mmol (equivalent to a salt reduction of 4.4 g/day of salt), the mean reduction in systolic BP was −5.39 mmHg (95% CI: −6.62 to −4.15), and the mean reduction in diastolic BP was −2.82 mmHg (95% CI: −3.54 to −2.11). Whereas, in normotensive individuals the median reduction in 24-h urinary sodium excretion was 75 mmol (−4.4 g/day of salt), the mean reduction in systolic blood pressure was −2.42 mmHg (95% CI: −3.56 to −1.29) mmHg, and the mean reduction in diastolic blood pressure was −1 mmHg (−1.85 to −0.15). Moreover, meta-regression showed a dose-response relationship between a reduction in salt intake and BP decrease in the long-term, after adjusting for age and BP status [47].

**Table 1 medicina-58-01175-t001:** Summary of the main evidence reported by meta-analyses regarding the effect of salt restriction upon the main cardiovascular and renal outcomes.

Outcome	Evidence from Meta-Analyses
BP	Dose-response relationship between salt reduction and BP decrease (He 2013 [47], Taylor 2011 [48], Graudal 2020 [32], Aburto 2013 [49])
CV events	High sodium intake increased risk of: Stroke Stroke mortality Coronary heart disease (Strazzullo 2009 [40], Aburto 2013 [49], Jayedi 2019 [45]) 2 Low sodium intake: Decreased risk of all-cause mortality and CHD (Khan 2019 [50]) Increased risk of all-cause mortality and CVD incidence compared to the usual sodium intake (<2.7 g/day versus 2.7–7 g/day)—U-shaped curve (Graudal 2014 [42]) No strong effect on all-cause mortality and CVD morbidity (Taylor 2011 [48])
HF	Severe salt restriction: Increased all-cause mortality (Taylor 2011 [48]) Slightly improved clinical parameters in outpatient setting, but inconclusive results in hospitalized patients (Mahtani 2018 [51])
Lipids	Salt restriction: Increase in total cholesterol and triglycerides with severe salt restriction (Graudal 2020 [32]) No effect with long-term moderate salt restriction (He 2013 [47])
Kidney function	Salt reduction: Decrease in urinary albumin excretion, more so in CKD patients (d’Elia 2015 [31]) Decrease in proteinuria and albuminuria in early CKD, CKD stages 1–4 and diabetic kidney disease (McMahon 2021 [52], Garofalo 2018 [53], Chen 2022 [54])

BP = blood pressure, CV = cardiovascular, CHD = coronary heart disease, CKD = chronic kidney disease, CVD = cardiovascular disease, HF = heart failure.

Compared to the Dietary Approaches to Stop Hypertension (DASH) diet, which lowers BP within 1 week with no significant changes further on, a low-sodium regular diet lowers BP up to 4 weeks and beyond [55]. Thus, the direct association between sodium intake and hypertension is not time limited.

#### 3.4.2. Salt Intake and CKD

It is clear that in order to maintain a good hemodynamic balance, it is necessary to have a good functional kidney. Healthy kidneys compensate for sodium and water excess according to body needs. However, the increase in extracellular volume becomes a major issue in patients with kidney dysfunction, especially in end-stage renal disease (ESRD) [56]. In dialysis patients, for which maintaining an extracellular fluid balance is crucial, the intracellular-extracellular water balance is dependent both on sodium intake during the interdialytic phase, and on sodium removal during hemodialysis sessions [57]. Most of the sodium is removed through convection and diffusive losses, depending on the sodium gradient between dialysate and plasma serum. Evidence suggests that patients have an individual, stable osmolar set point; thus, using fixed dialysate sodium concentrations leads to a positive sodium balance, which could lead to increased thirst, interdialytic weight gain and high BP in salt-sensitive individuals [57].

However, it is also fundamental to specify that there are different mechanisms by which excessive salt intake is believed to cause health problems, and does not necessarily involve BP elevation. Evidence shows that besides increasing vascular resistance, excessive salt ingestion induces hypertrophy of vascular muscle cells, cardiac hypertrophy, and production of reactive oxygen species [58,59]. More so, excessive salt intake is associated with arteriosclerosis, arteriolosclerosis, high oxidative stress, renal parenchymal damage and also determines a decrease in life span [25,60,61]. For instance, Yu et al. [62] demonstrated that the administration of 8.0% salt diet to lab rats conducted to TGF1 production and fibrosis in both the kidney and the left ventricle, concluding that salt intake would play an important role in cardiovascular and renal fibrosis and dysfunction [62].

The beneficial effects of salt reduction are well confirmed by dedicated meta-analyses: moderate salt restriction significantly reduces proteinuria in CKD stage 1–4 and diabetic kidney disease (Table 1) [52,53,54].

#### 3.4.3. Salt Intake and Stroke

It is important to specify that the health benefits of a reduction in salt intake are generally explained by the decrease in BP values. However, the relationship between high sodium intake and arterial stiffness appears independent of blood pressure values, as pulse-wave velocities are lower in low-salt consumers compared to BP-matched controls [63]. While initial meta-analyses [40,45,49] reported an increased stroke risk in high-salt diets (see above), a recent umbrella review of meta-analyses [50] found no association between diet sodium and stroke risk. however, it reported moderate certainty beneficial effects of a low-salt diet both in normotensive and hypertensive individuals, where it is associated with decreased all-cause mortality and reduced incidence of coronary heart disease (Table 1). Specifically, hypertensive patients would also benefit from a decreased cardiovascular mortality with reduced sodium intake [50].

#### 3.4.4. Salt Intake and HF

At the end of the pathogenic vascular dysfunction related to a high-salt diet lies the development of HF. Not only the elevated BP values related to an increased sodium intake, but also arterial stiffness, oxidative stress and systemic inflammation. Increased renin and aldosterone action all promote cardiac hypertrophy and fibrosis, resulting in HF. The DASH diet, which limits foods that are rich in sodium, is useful in the primary prevention of HF. Sodium restriction is also appropriate for the early stages of HF (“at risk” or “asymptomatic”). Finally, sodium restriction is thought to improve the outcomes of class III to IV HF by reducing BP values, the levels of B-type natriuretic peptide, aldosterone, plasma renin activity, pulmonary capillary wedge pressure and oxidative stress [64].

Nevertheless, it seems that moderate sodium restriction up to 2.8 g/day is even more beneficial in HF, compared to low sodium restriction of up to 1.8 g/day [64]. Recent reviews showed that a dietary sodium restriction of 2.6–3 g/day is effective in mitigating the negative neurohormonal pathogenesis of HF [65]. This adds to pre-existing data showing that low sodium restriction (<3 g/day) is associated with worse cardiovascular prognosis compared to moderate sodium consumption [66]. Below, we discuss the side-effects of a low-salt diet.

### 3.5. “Salt Not Always an Enemy”: The Deleterious Effects of Salt Restriction

#### 3.5.1. CVD and All-Cause Mortality

An increasing number of studies regarding sodium impact on health do not support the general benefits associated with salt restriction upon CVD and related events (Table 1). For example, a Cochrane meta-analysis published by Taylor et al. [48] analyzed the effect of salt reduction in seven different studies that included normotensive, hypertensive and HF patients. The authors found that by reducing salt intake in these patients, there was no strong effect in declining the risk of all-cause mortality and CVD morbidity, despite reducing BP levels. Furthermore, in patients with HF, salt restriction was associated with an increased risk of all-cause mortality [48]. The differences between the reported results among literature may be due to inconsistencies regarding study design and also patients’ lifestyle: subjects willing to follow a low-salt diet are more prone to following a healthy lifestyle that may contribute to improved cardiovascular outcomes, while experimentally allocating a low-sodium diet may differently impact the patient’s adhesion and the outcomes of the study [40,48].

Additionally, other large studies also failed to show the benefits of dietary sodium restriction upon CVD, or even demonstrated adverse effects related to sodium restriction. Data from the second National Health and Nutrition Examination Survey (NHANES II study) showed that a sodium restriction of less than 2.3 g/day is associated with a higher incident CVD and all-cause mortality compared to a sodium ingestion more than 2.3 g/day [67]. Conversely, NHANES III study found no significant relationship between a higher sodium intake and CVD or ischemic heart disease mortality, but a significant monotonic association between an increased sodium/potassium ratio and CVD mortality [68]. Thus, the curvilinear association between sodium intake and cardiovascular events found in some studies may be explained, at least partially, by the co-existence of a low potassium diet. A recent review endorsed by the European Society of Cardiology concludes that the high cardiovascular risk associated with an increased sodium diet may be mitigated by a potassium-rich diet (fruits, vegetables and nuts) [10].

#### 3.5.2. HF

In addition, in patients suffering from congestive HF, a normal-sodium diet could be preferred. Studies show that low-sodium diets are associated with an increase in renin, aldosterone, noradrenaline and BNP and also an increase in plasma renin activity, compared to a normal-sodium diet in patients with compensated congestive HF [69]. This leads to vascular congestion that promotes a pro-oxidant, pro-inflammatory gene expression in endothelial cells, thus promoting cardio-renal dysfunction [64]. It also seems that besides a beneficial hormonal profile linked to a normal-sodium diet, these patients also experience better clinical outcomes [69]. Recent data from randomized clinical trials showed that aggressive sodium and fluid restriction in decompensated HF with preserved ejection fraction does not reduce readmission rates or mortality. It is associated with similar degrees of decongestion when compared to a normal sodium diet [51,70,71]. In a recent systematic review performed by Mahtani et al. [51], the effects of dietary salt restriction in hospitalized patients was inconclusive, while it seemed to be associated with slightly improved clinical parameters among outpatients (Table 1) [51]. However, there was a paucity of data regarding the effects of salt restriction upon CV, all-cause mortality and adverse events (such as myocardial infarction or stroke) in HF patients [51]. As such, the latest European guidelines in 2021 recommend “avoiding excessive salt intake (>5 g/day)” [14].

#### 3.5.3. Metabolic Parameters

In their meta-analysis concerning salt reduction towards the deleterious effects of severe sodium restriction upon metabolic parameters, Graudal et al. [32] have drawn attention towards an increase in serum lipids (total cholesterol and triglycerides), especially in people with normal BP. This might actually mitigate the beneficial effects upon BP when evaluating the overall CV risk, thus explaining the U-or J-shaped association between salt restriction and CV events. However, the meta-analysis of He et al. [47] didn’t find any harmful effects of long-term modest salt reduction upon lipid metabolism (Table 1) [47].

#### 3.5.4. Hypotension

Nevertheless, the cardiovascular benefits of sodium restriction may be explained by the BP-lowering effects that resulted as a consequence of a large reduction in salt intake in hypertensive patients, hence the small cardiovascular advantage observed in some studies [72]. On the other hand, sodium represents an important mineral which is necessary in the maintenance of plasma volume, acid-base balance, transmission of nerve impulses, and normal cell function [49]. Considering the conflicting evidence, it is wise to be cautious in prescribing salt restriction to patients. It could be useful to carefully choose patients who would benefit from dietary salt restrain (Figure 1).

Moreover, sodium intake is believed to be involved in the pathogenesis of hypertension, hypervolemia, and mortality in hemodialysis patients. That is why sodium restriction is almost universally recommended. However, some studies showed that elevated blood pressure is not always a negative aspect. For instance, in patients with chronic hemodialysis, arterial hypertension is associated with better survival rates compared to hypotension (“reverse epidemiology”). Hypotension in dialyzed patients increases mortality, while being associated with serious vascular complications. These include cerebral infarction, cardiac or mesenteric ischemia, and also symptoms such as muscle cramps, abdominal and chest pain, nausea, vomiting, dyspnea, light-headedness, weakness, anxiety, vertigo, paleness and sweating, all of which negatively impact the patient’s quality of life [73]. Despite both hypertension and hypotension being undesirable in dialysis patients, hypotension could represent a more dangerous situation for this category, given the fact that the compensatory mechanisms are hampered by the kidney dysfunction [74].

## 4. Conclusions

While excessive sodium intake is a public health concern, current evidence is truly heterogenous with regards to the extent of optimal sodium reduction to be implemented in the general population. Studies evaluating sodium intake in the higher quartile compared to the lower quartile reported a higher incidence of cardiovascular events related to increased diet sodium. However, when comparing moderate sodium consumption to low sodium restriction, moderate is better in the general population. While the beneficial effects in reducing BP levels are confirmed, a moderate or normal sodium consumption (3 to 6 g/day) appears optimal with regards to reducing cardiovascular risk, compared to both low and high-sodium diets. Even HF patients are advised not to increase their diet sodium, rather than to aggressively restrict its use.

### Perspectives

While hypertensive patients, older subjects or those having low-potassium diets could benefit more from salt restriction, a low-sodium diet could do more harm than good in certain patient categories. Even in the first category of patients, randomized controlled trials are still awaited to establish the effectiveness and safety of optimal cut-offs for salt restriction regarding cardiovascular health. Avoiding excessive sodium intake (>6 g/day) is generally demonstrated as beneficial and recommended at the populational level; however, endorsing severe sodium restriction should be tailored upon each patient, according to their comorbidities, especially as consensus between various guidelines and studies on this topic are lacking.

## Figures and Tables

**Figure 1 medicina-58-01175-f001:**
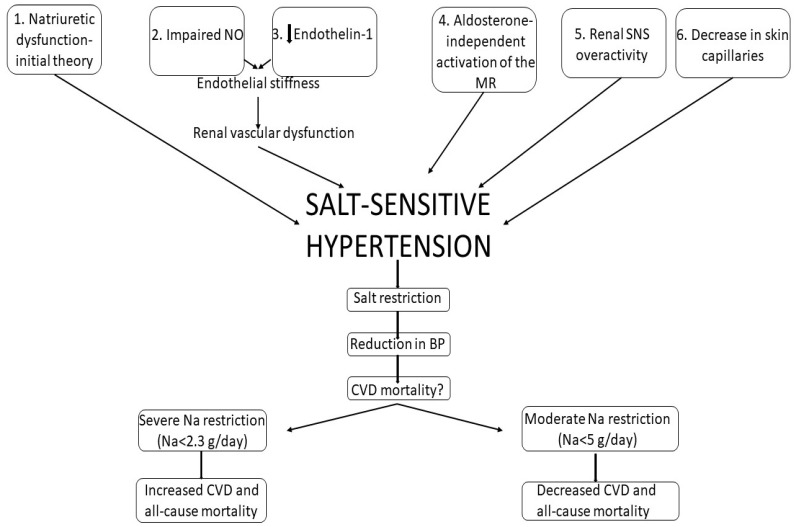
Development of salt-sensitive hypertension and various consequences of salt restriction. BP = blood pressure, CVD = cardiovascular disease, MR = mineralocorticoid receptor, Na = sodium, NO = nitric oxide, SNS = sympathetic nervous system.

## Data Availability

All data are available on a simple request.

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
