# Peer review of "Salt, Not Always a Cardiovascular Enemy? A Mini-Review and Modern Perspective"

_medicina, 2022, doi:10.3390/medicina58091175_

Round 1

Reviewer 1 Report

The topic of this review is very interesting for research and public health.

They pointed out the unfavorable cardiovascular effects and few controversial data in particular settings of the high salt intake.

However, the authors reported several data that may generate confusion in this context. Because reported both data for general population and selected sample of patients (e.g. with heart failure or kidney disease).

The key message is very hazardous and does not support the international recommendations and strategy to reduce daily salt intake in particular at population level, that are supported by a large bulk of evidence. 

Furthermore, high salt intake has detrimental effects also independently of blood pressure levels, indeed it is involved in the arterial stiffness damage (PMID: 29084085 ) or in the kidney damage (microalbuminuriaPMID: 26240299  ). Of note, high salt intake is also well recognized factor associated with development of gastric cancer (PMID: 22296873 ).

Finally, the sentence stated by authors that hypertension is an advantage is very hazardous in this context and only speculative. 

Author Response

Reviewer 1

The topic of this review is very interesting for research and public health.

They pointed out the unfavorable cardiovascular effects and few controversial data in particular settings of the high salt intake.

We kindly thank the reviewer for the comment.

However, the authors reported several data that may generate confusion in this context. Because reported both data for general population and selected sample of patients (e.g. with heart failure or kidney disease).

The manuscript presents data regarding the impact of salt intake in the general population from a cardiovascular health point of view. Heart failure and kidney disease are two conditions particularly associated with sodium imbalance, often interconnected, and where changes in sodium intake can drastically alter the systemic and renal hemodynamics, respectively. Therefore, we chose to discuss these selected conditions as well. We amended the structure of the manuscript in order to make it more clear: subheadings anticipating the specific referral to HF and CKD were added.

The key message is very hazardous and does not support the international recommendations and strategy to reduce daily salt intake in particular at population level, that are supported by a large bulk of evidence. 

We fully agree with the reviewer and made the key message clearer in the end of the manuscript: avoiding excessive salt intake is beneficial and recommended at populational level. However, endorsing severe sodium restriction should be tailored upon each patient, according to their comorbidities, especially as consensus between various guidelines and studies on this topic are lacking.

Furthermore, high salt intake has detrimental effects also independently of blood pressure levels, indeed it is involved in the arterial stiffness damage (PMID: 29084085 ) or in the kidney damage (microalbuminuriaPMID: 26240299  ). Of note, high salt intake is also well recognized factor associated with development of gastric cancer (PMID: 22296873 ).

We thank the reviewer for the valuable pieces of information, we added the indicated references in the manuscript.

Finally, the sentence stated by authors that hypertension is an advantage is very hazardous in this context and only speculative. 

We kindly thank the reviewer for the observation, the sentence was modified accordingly to clarify its meaning: “in patients with chronic hemodialysis, arterial hypertension is associated with better survival rates compared to hypotension (“reverse epidemiology”)” instead of “is an advantage”.

Reviewer 2 Report

The topic covered in this article is interesting . However, I have some concerns:

1. Based on what criteria did you select the articles in the study?

2. You could improve the structure of the article: introduction, materials and methods, discussion, conclusions ( limitations / new perspectives).

3. More pictures, tables and figures might improve the article.

Author Response

Reviewer 2

The topic covered in this article is interesting . However, I have some concerns:

  1. Based on what criteria did you select the articles in the study?

A description of the Methods used for the writing of this paper was added in the manuscript.

  1. You could improve the structure of the article: introduction, materials and methods, discussion, conclusions ( limitations / new perspectives).

We kindly thank the reviewer for the suggestion; we improved the structure of the article, accordingly.

  1. More pictures, tables and figures might improve the article.

Table 1 summarizing current evidence from meta-analyses regarding the impact of salt reduction upon CV and kidney outcomes was added.

Reviewer 3 Report

Dear Authors,

This manuscript is a mini review discusses current literature data regarding the advantages and disadvantages of salt restriction and how it impacts the overall health status. The study presents an updated theoretical framework. The theme is applied from a health point of view.

General concept comments

1)      In topic “Salt not always an enemy: the deleterious effects of salt restriction”: I suggest that the authors include information on the possible influences of salt restriction on metabolic variables such as LDL, cholesterol, triglycerides, and insulin resistance.

Specific comments

1)      I suggest updating the reference number 22: DOI: 10.1002/14651858.CD004022.pub5

2)      It is necessary to review the formatting of the references according to the journal's rules.

3) There are references with missing information and with incorrect information.

Author Response

Reviewer 3

Dear Authors,

This manuscript is a mini review discusses current literature data regarding the advantages and disadvantages of salt restriction and how it impacts the overall health status. The study presents an updated theoretical framework. The theme is applied from a health point of view.

General concept comments

  • In topic “Salt not always an enemy: the deleterious effects of salt restriction”: I suggest that the authors include information on the possible influences of salt restriction on metabolic variables such as LDL, cholesterol, triglycerides, and insulin resistance.

A subheading and corresponding paragraph regarding the effect of salt restriction upon metabolic parameters.

Specific comments

  • I suggest updating the reference number 22: DOI: 10.1002/14651858.CD004022.pub5

We kindly thank the reviewer for the suggestion, we updated the above-mentioned reference.

  • It is necessary to review the formatting of the references according to the journal's rules.

The format of the references was updated according to journal’s rules. Thank you!

  • There are references with missing information and with incorrect information.

References were extensively reviewed and updated. Thank you!

Reviewer 4 Report

Dear authors
Kindly please see my comments attached. All the best. 

Author Response

  1. Salt, not always a cardiovascular enemy? A mini-review and modern perspective

When “mini-review” is put, I expect to see 1 section for the methodology of the mini-review. Which no methodology is stated. The evidence use should be higher level of evidence such as metaanalysis → SR → RCT → clinical guidelines (at least) E.g; keywords search terms used, databases used, and the conduction of literature searches

We kindly thank the reviewer for the comment, we added the Methods section describing the methodology of research that led to the writing of this manuscript.

  1. Abstract: pg 1 salt-sensitive arterial hypertension. Is this based on ICD 10 code classifications? Or any specific definition? Proposed definition for this. Common used term is saltsensitive hypertension.

We kindly thank the reviewer for the observation, the terminology error was corrected.

  1. Abstract: pg 1 Excessive salt intake leads to arterial stiffness The direct pathophysiology is salt increase arterial pressure. Perhaps rephrase to avoid confusion (please check on evidence).

While it is true that high BP leads to arterial stiffness, evidence shows that an increased sodium loading directly impacts the endothelium via the upregulation of endothelin 1 and increased production of TGF-β1, and this is not necessarily dependent on BP changes. Therefore, both are true (salt increases BP, one mechanisms being the increase in arterial stiffness) (Ertuglu Front Physiol 2021, Du Pont J Hypertens 2013, Sanders Am J Physiol-Ren Physiol 2013).

  1. Abstract: pg 1 Excessive salt intake leads to arterial stiffness in susceptible individuals via impaired nitric oxide action and increased endothelin1 expression, overactivity of the renal sympathetic nervous system and also via aldosteroneindependent activation of the mineralocorticoid receptor. Again the direct pathophysiology is CKD itself increased endothelin, hence increase nitrous oxide -→ arterial stiffening. Hence need rephrase to avoid confusion.

We thank the reviewer for the comment. Probably the reviewer was referring to BP, instead of CKD. While we agree that arterial stiffness is a consequence of high BP, experimental data demonstrated the other way around as well: endothelin production is increased in response to high salt intake independently in a blood pressure-independent manner (Tsai Exp Biol and Med 2006, Speed FASEB J 2015, and also thr above-mentioned references).

  1. Abstract: pg 1 Whether this translates into improved cardiovascular outcomes is still to be proven. This referring to what? If salt/ arterial stiffness is well established knowledge. Salt reduction → reduce BP → reduce CVD TOC

We made the phrase clearer. We were referring to the J-shaped type of relationship between salt restriction and the incidence of cardiovascular events or mortality: to what extent is sodium restriction beneficial? Meta-analyses reported a lower incidence of CVD in moderate sodium intake diets compared to a more severe restriction (<2.7 g/day) (Graudal Am J Hypertens 2014). However, data are still controversial.

  1. Abstract: pg 1 salt-resistant hypertension Is this based on ICD 10 code classifications? Or any specific definition? Proposed definition for this. Common used term is saltsensitive hypertension.

Salt-resistant hypertension is a long-used term generally referring to arterial hypertension that does not meet the criteria for salt-sensitive hypertension (rise in BP values by at least 5% after sodium loading and decrease in BP after sodium depletion) (de la Sierra Clin Sci (Lond) 1995, Mishra Indian Heart Journal 2018). The terminology in the abstract was amended in order to eliminate confusion. 

  1. Keywords Keywords: salt; arterial hypertension; salt restriction If you’ve came across the terms/definition of “salt-sensitive” or “salt-resistant”, it should be in your keywords

We kindly thank the reviewer for the comment, the keywords were amended.

  1. Introduction The subject of dietary salt has been for a long time a focus of clinical research and is still representing a subject of intense study and debate. The special interest upon salt dietary arises from the fact that sodium could represent a relatively easy to control health risk factor, among other exogenous risk factors such as tobacco, alcohol, or caloric intake. Please add reference for this statement.

We added the indicated reference.

  1. Introduction Thus, one may assume that salt restriction may have several health benefits in the general population. Please add what evidence says on this statement.

Evidence on health benefits of salt reduction in the general population were added.

  1. Introduction According to salt hypothesis, reducing the amount of salt intake will result in a decrease of BP levels and in a subsequent reduction of the cardiovascular events [2]. In scientific evidence, please quantify the definition of reduce amount of salt. E.g: recommendation in CPG HPT - of 1 teaspoon/ 5gm salt per day (this is evident).

We thank the reviewer for the observation. We added the amount of salt reduction that statement was referring to, according to the data cited. The recommendation of CPG for HPT are already given below, further-on in the manuscript.

  1. Introduction This applies to salt-sensitive hypertension. However, there is an inter-individual variation to the blood pressure raising effects of sodium charge, with a specific category of patients having salt-resistant hypertension [3]. Please define salt-sensitive hypertension & salt-resistant hypertension. This is very important if new terms are introduced. Simple phrasing to define, before you elaborate in page 2 (the theories).

Salt-sensitive and salt-resistant hypertension were defined, accordingly.

  1. Page 2 Current recommendations for salt intake I would suggest separate the message based on 3 points: i. salt requirement recommendations for general population ii. salt restriction recommendations for HF patients iii. salt restriction recommendations for CKD patients. iv.

We structured the paragraph and the message, accordingly.

  1. Page 2 An extreme sodium restriction is discouraged since it was proven that a very low sodium diet is associated with negative health consequences due to competing “off-target” adverse consequences upon activation of the sympathetic response and renin-angiotensin-aldosterone (RAS) system, that increased the risk for cardiovascular events [4]. This led to paradigm shifts in the official recommendations for sodium intake in HF patients, where the latest American (2022) and European (2021) guidelines recommend avoiding excessive salt intake, rather than setting cut-offs for sodium restriction [7,8]. This sentences are confusing. What is the message here? The message here is “sodium restriction” or “avoid excessive salt intake” versus “cut off sodium restrictions”.

We amended the paragraph in order to clarify the message. The message for the general population is to avoid excessive salt intake, with different targets endorsed by the WHO and The ESC, while sodium restriction below 5 g/day is recommended for patients with a high CV risk, HF or CKD.

  1. Page 2 As a more consistent reduction in sodium intake close to the lower range suggested by the American Heart Association is difficult to achieve and sustain on the long-term, it may still be beneficial in particular cases, such as salt-sensitive hypertensive patients. Reference for this?

The phrase was amended and the corresponding reference was inserted.

  1. Page 2 Salt-sensitive versus salt-resistant hypertension Ok, this is good.

We kindly thank the reviewer for the comment.

  1. Page 3 For instance, a study published in 2011 on young and adult uninephrectomized rats fed with highsalt diet for four weeks showed a significant increase in BP (180.9 ± 22.6 mmHg) and proteinuria (100.3 ± 44.5 mg/day), as compared to normal sodium fed rats (BP 117.8 ± 9.5 mmHg, proteinuria 8.8 ± 1.2 mg/day). However, the adult high-salt fed rats showed only a moderate increase in BP, when compared to younger rats and no increase in proteinuria level. Do you have evidence on human?

Evidence from clinical studies was added and the paragraph was amended.

  1. Page 4 Salt and cardiovascular events Again the terms used, as comments above. As arterial hypertension

We  corrected the terminology.

  1. Page 4 are inconsistent Based on ref 30-32, the words inconsistent maybe replaced by phrasing that there is range of evident salt intake

We replaced the phrasing: “inconsistent” was replaced with “ the range of results regarding the effect of increased diet sodium on all-cause mortality and all CVD is quite wide”.

  1. Page 5 Low-salt diet - advantages Suggest subheading: Low salt diet and blood pressure.

Subheading was added.

  1. Page 5 & 6 To arrange & add subheadings: 1. Low salt diet & BP 2. Salt intake & CKD 3. Salt intake & HF 4. Salt intake & stroke

The paragraph “Low salt diet-advantages” was sub-divided and subheadings were added.

  1. Page 6 “Salt not always an enemy”: the deleterious effects of salt restriction Subheadings of: 1. CVD & all cause mortality 2. HF 3. Hypotension

We added the indicated subheadings.

  1. Page 7 Perspective there is currently insufficient proof Suggest to use other word than insufficient. There were numerous studies, Perhaps the need to really go for meta-analyses on each subheadings suggested. Cochrane has rigorous metaanalyses and the quality is good. So these go back to the methodology of the mini-review. The conclusion reflects by the rigorous methodology done.

We kindly thank the reviewer for the comment, it definitely led to the improvement of the manuscript. The phrase in the end regarding “insufficient proof” was amended. Also, latest meta-analyses regarding the effect of salt restriction upon each outcome (BP, CKD, HF, CV events, metabolic parameters) were added, accordingly.

Round 2

Reviewer 1 Report

-

Author Response

Thank you for the kind corrections which help us improve our manuscript.